# Isolation and Functional Characterization of Two *SHORT VEGETATIVE PHASE* Homologous Genes from Mango

**DOI:** 10.3390/ijms22189802

**Published:** 2021-09-10

**Authors:** Xiao Mo, Cong Luo, Haixia Yu, Jinwen Chen, Yuan Liu, Xiaojie Xie, Zhiyi Fan, Xinhua He

**Affiliations:** State Key Laboratory for Conservation and Utilization of Subtropical Agro-Bioresources, College of Agriculture, Guangxi University, 100 East Daxue Road, Nanning 530004, China; 18376683951@163.com (X.M.); 22003luocong@gxu.edu.cn (C.L.); yuhaixia0201@163.com (H.Y.); m19977680073@163.com (J.C.); 18638785643@163.com (Y.L.); xiexiaojie0827@126.com (X.X.); 18178878668@163.com (Z.F.)

**Keywords:** mango, *SVP*, flowering time, expression analysis, functional analysis

## Abstract

The *SHORT VEGETATIVE PHASE (SVP)* gene is a transcription factor that integrates flowering signals and plays an important role in the regulation of flowering time in many plants. In this study, two full-length cDNA sequences of *SVP* homologous genes—*MiSVP1* and *MiSVP2*—were obtained from ‘SiJiMi’ mango. Sequence analysis showed that the *MiSVPs* had typical MADS-box domains and were highly conserved between each other. The analysis of expression patterns showed that the *MiSVPs* were expressed during flower development and highly expressed in vegetative tissues, with low expression in flowers/buds. The *MiSVPs* could responded to low temperature, NaCl, and PEG treatment. Subcellular localization revealed that MiSVP1 and MiSVP2 were localized in the nucleus. Transformation of *Arabidopsis* revealed that overexpression of *MiSVP1* delayed flowering time, overexpression of *MiSVP2* accelerated flowering time, and neither *MiSVP1* nor *MiSVP2* had an effect on the number of rosette leaves. Overexpression of *MiSVP1* increased the expression of *AtFLC* and decreased the expression of *AtFT* and *AtSOC1*, and overexpression of *MiSVP2* increased the expression levels of *AtSOC1* and *AtFT* and decreased the expression levels of *AtFLC*. Point-to-point and bimolecular fluorescence complementation (BiFC) assays showed that MiSVP1 and MiSVP2 could interact with SEP1-1, SOC1D, and AP1-2. These results suggest that *MiSVP1* and *MiSVP2* may play a significant roles in the flowering process of mango.

## 1. Introduction

Flower transformation is the transition from vegetative growth to reproductive growth and is an important stage of the plant life cycle. Plant flowering is a hotspot in botanical research [1]. This process is jointly affected by environmental conditions and the internal growth and developmental state. In *Arabidopsis thaliana*, flowering regulation consists of six signaling pathways: the photoperiod, age, ambient temperature, gibberellin, vernalization pathways and the autonomous pathway comprising a combination of epigenetic factors and post-transcriptional gene regulation [2,3,4,5,6,7]. These regulatory pathways are independent and interact with each other to form a complex gene regulatory network to jointly regulate plant flowering [8].

*SHORT VEGETATIVE PHASE (SVP)* homologous genes belonging to the StMADS11 group of the MADS-box gene family are critical flowering repressors in *Arabidopsis* and have important functions in regulating floral transition and inflorescence structure in many plants [9]. *SVP* is regulated by autonomous and thermosensitive gibberellin pathways [10]. Recent research has suggested that *SVPs* may be another central regulators of the flowering regulatory network because *SVPs* can repress *SUPPRESSOR OF OVEREXPRESSION OF CONSTANS1 (SOC1)*, *FLOWERING LOCUS T (FT)*, and *TWIN SISTER OF FT (TSF)* transcription in the meristem and leaf [10,11]. *SVP* can also bind to the promoter regions of *FT* and *SOC1*, and *SVP* can negatively regulate the expression of *FT* by binding to the CArG motif in the *FT* promoter sequence, which is one of the molecular mechanisms regulating flowering time under the condition of temperature fluctuations [12]. SVP can also interact with FLM-β in response to the regulation of ambient temperature, and *SVP* is an important mediator within the thermosensitive pathway [13]. In the early stages of flower development, *AGAMOUS-LIKE 24 (AGL24)*, *SVP*, and *APETALA1* (*AP1*) form the dimers AP1-AGL24 and AP1-SVP, which interact with the LEUNIG (LUG)-SEUSS (SEU) compound to regulate the expression of *AGAMOUS (AG)*, thereby affecting the construction of flower organs [14].

Mango (*Mangifera indica* L.) is an economically important tropical fruit and is cultivated in many countries. The flowering and fruit setting of mango are important factors affecting its production. Studies have shown that mango flowering is affected by many factors, including drought, low temperature, age of branches, and gibberellin and paclobutrazole treatment [15]. Some flowering-related genes of mango, such as *MiAP1*, *MiSOC1*, *MiFT*, and *MiCO,* have been isolated and functionally identified [16,17,18,19]. However, the functions of the *SVP* genes in mango have not been reported. In this study, two *SVP* homologue genes were isolated, and their functions were studied in transgenic *A. thaliana*.

## 2. Results

### 2.1. Cloning and Bioinformatics Analysis of MiSVPs

Two *MiSVP* genes, named *MiSVP1* (MZ542518) and *MiSVP2* (MZ542519), were identified from transcriptomic and genomic data (unpublished data) of ‘SiJiMi’ mango. The DNA sequences of *MiSVP1* and *MiSVP2* were 7442 bp and 6305 bp, respectively, in length, and included eight exons and seven introns (Figure 1A). The open reading frames (ORFs) were 675 bp and 657 bp and encoded 225 and 219 amino acids (aa), respectively. Their sequences were highly similar to each other. The molecular weight (MW) of MiSVP1 and MiSVP2 was estimated to be 25.55 and 24.77 kDa, respectively, and the theoretical isoelectric point (pI) of MiSVP1 and MiSVP2 was estimated to be 6.33 and 6.58, respectively. Sequence analysis indicated that the two MiSVP proteins contained typical MADS-box and K-box domains (Figure 1B). Phylogenetic tree analysis showed that MiSVP1 and MiSVP2 were clustered within the same branch and closely related to *Pistacia vera* PvSVP (XP_031287661) (Figure 1C).

### 2.2. Expression Analysis of MiSVPs

#### 2.2.1. Tissue Expression Analysis of MiSVPs

To research the function of *MiSVP1* and *MiSVP2* in the growth and development of mango, the expression levels of *MiSVP1* and *MiSVP2* in all tissues of ‘SiJiMi’—including the nonflowering branches of leaves, stems, buds, and flowering branches of leaves, stems, and flowers—were investigated. The real-time quantitative reverse transcription polymerase chain reaction (qRT-PCR) results indicated that the *MiSVP1* and *MiSVP2* genes were expressed in all tested tissues (Figure 2A), and both *MiSVP* genes were more highly expressed in the leaves of flowering branches than in nonflowering branch tissues. The lowest expression level of *MiSVP1* was found in flowers, and the lowest expression of *MiSVP2* was found in buds. *MiSVP2* was more highly expressed than *MiSVP1* in all tested tissues.

#### 2.2.2. Temporal Expression Analysis of MiSVPs

To further study the role of *MiSVP1* and *MiSVP2* in mango, the expression levels of *MiSVP1* and *MiSVP2* in stems, leaves and buds from November 2018 to March 2019 were analyzed (Figure 2B,C). The qRT-PCR results showed that *MiSVP1* and *MiSVP2* were present in samples at all stages of the test phase but at different transcriptional levels. In leaves, the expression of *MiSVP1* was highest in the vegetative growth phase, followed by the early stage of floral induction and the period of floral differentiation. The expression level of *MiSVP1* was lowest in the early stage of floral differentiation, at which point the inflorescences had begun to differentiate. The expression level of *MiSVP1* in stems was highest in the early stage of floral differentiation, followed by the vegetative growth phase and the early stage of floral induction, and the lowest expression levels were observed during the late stage of floral differentiation. The expression level of *MiSVP1* in buds was highest in the early stage of floral induction, followed by the vegetative growth phase and the late period of floral differentiation, and the lowest expression levels were observed during inflorescence elongation and flowering. *MiSVP2* had a different expression pattern than *MiSVP1.* The expression level of *MiSVP2* in leaves and buds showed a trend of first decreasing and then increasing, and the lowest expression levels occurred during floral differentiation. The expression level of *MiSVP2* in stems was highest in the early stage of floral differentiation, followed by the late period of floral differentiation and the vegetative growth phase, and the lowest expression levels were observed during the early stage of floral induction. The expression level of *MiSVP2* was higher than that of *MiSVP1* from 5 November to 6 March.

#### 2.2.3. Effects of Adversity Treatments on the Alteration of MiSVPs Expression Levels

To explore the impact of adversity on the alteration of *MiSVPs* expression level, the expression levels of *MiSVPs* were detected after one of several stress treatments. Low temperature treatment significantly upregulated the expression levels of *MiSVP1* and *MiSVP2* compared with those at 0 h, with expression peaking at 48 h after treatment (Figure 3A). After 30% PEG treatment, *MiSVP1* and *MiSVP2* were significantly upregulated compared with 0 h levels and reached peak expression at 6 h (*MiSVP2*) and 48 h (*MiSVP1*) after treatment (Figure 3B). After salt stress treatment, the expression of *MiSVP1* was significantly downregulated at 6 h; then upregulated, reaching a peak at 12 h; and finally downregulated again. The expression of *MiSVP2* was significantly downregulated at 6 h, returned to the pretreatment level between 12–24 h, and then was significantly upregulated at 48–72 h (Figure 3C).

### 2.3. Subcellular Localization of MiSVPs

To ascertain the specific locations of MiSVP proteins in the cell, 35S::MiSVP1-GFP and 35S::MiSVP2-GFP fusion protein carriers were transferred into *Agrobacterium tumefaciens* (EHA105) and then injected into tobacco leaves to observe the fluorescence signal. The green fluorescence of 35S::GFP was used as a control and could be found in the cell membrane and nucleus, while the green fluorescence of the 35S::MiSVP1-GFP and 35S::MiSVP2-GFP fusion proteins were only observed in the nucleus (Figure 4), which showed that the MiSVP1 and MiSVP2 proteins were located in the nucleus.

### 2.4. Functional Analysis of MiSVPs in Arabidopsis thaliana

To study the role of the two *MiSVP* genes in plant flowering, the *MiSVPs* genes were transfected into wild-type (WT) *A. thaliana*. Three homozygous transgenic lines of *MiSVP1* (OE-1#2, OE-1#3, and OE-1#6) and of *MiSVP2* (OE-2#5, OE-2#8, and OE-2#12) were selected for functional analysis. As shown in Figure 5A,B, overexpression of *MiSVP1* delayed flowering time. However, overexpression of *MiSVP2* advanced flowering time. Compared with those in the WT, the expression levels of the *MiSVP1* and *MiSVP2* genes in transgenic plants were high (Figure 5C,D). Overexpression of the *MiSVPs* resulted in no significant changes in the number of rosette leaves (Figure 5E,F) but caused significant changes in days to initial flowering (Figure 5G,H).

To analyze the influence of overexpression of *MiSVPs* on the expression of endogenous genes in *A. thaliana*, leaves of transgenic plants were collected before blooming, RNA was extracted, and the expression levels of endogenous genes associated with flowering were analyzed by qRT-PCR. The expression of the endogenous genes *AtFT* and *AtSOC1* decreased in *MiSVP1* transgenic plants compared with WT, while that of the endogenous gene *AtFLC* increased (Figure 6A). The expression trends of endogenous genes in *MiSVP2* transgenic plants were opposite those in *MiSVP1* transgenic plants. In *MiSVP2* transgenic plants compared with WT, the expression of the endogenous genes *AtFT* and *AtSOC1* increased, while that of the endogenous gene *AtFLC* decreased (Figure 6B).

### 2.5. Screening and Validation of MiSVPs Interacting Proteins

To investigate the regulation of *MiSVPs* in the flowering network, the non-self-activated yeast pGADT7-MiSVP vector was combined with the mango cDNA library previously constructed in our laboratory, and a total of 17 blue monoclonal colonies were obtained by screening on selective media (SDO/–Trp/–Leu/–His/–Ade). Due to the high false positive rate of the yeast two-hybrid system, three flowering-related proteins were selected for point-to-point and bimolecular fluorescence complementation (BiFC) assays. These three proteins were SUPPRESSOR OF OVEREXPRESSION OF CONSTANS 1 D (SOC1D), APETALA 1-2 (AP1-2), and SEPALLATA 1-1 (SEP1-1). As shown in Figure 7, the point-to-point and BiFC assays of the three candidate proteins with MiSVPs showed that MiSVP1 and MiSVP2 interacted with SEP1-1, AP1-2 and SOC1D (Figure 7A,C).

## 3. Discussion

Different members of the MADS-box gene family play a wide range of roles in the regulation of flowering time, the determination of floral meristem and floral organ characteristics, and the regulation of root, leaf, ovule and fruit development [20]. *SVP* belongs to the STMADS11 group of the MADS-box gene family. In *A. thaliana*, *SVP* regulates flowering time and determines floral meristem and floral organ characteristics [9,21]. Recent advances in research on *SVPs* suggest that *SVP* may be another central regulator of the flowering regulatory network [10]. In this study, we isolated and identified two *SVP* homologous genes, *MiSVP1* and *MiSVP2,* from mango. The exon-intron structural analysis of *MiSVP1* and *MiSVP2* suggests that the intron-exon length and the intron number differ between *MiSVP1* and *MiSVP2*. There were seven introns from the start codon to the stop codon of *MiSVP1* and *MiSVP2* (Figure 2A), which is the same number as in *Prunus mume* and different from that in *A. thaliana,* with eight introns in *Poncirus trifoliata L.* Raf. [22,23]. These results indicated that *SVP* gene sequences differ among different species. Sequence analysis indicated that both MiSVP1 and MiSVP2 belonged to the MADS-box gene family, with typical MADS-box motifs and high conservatism between them (Figure 2B). Phylogenetic tree analysis showed that MiSVP1 and MiSVP2 belong to the same branch, but that MiSVP2 is more closely related than MiSVP1 to the *SVP*-like gene PvSVP of *Pistacia vera* (Figure 2C), which indicated that MiSVP2 and PvSVP are very closely related; their protein structures may also be very similar. Therefore, it is speculated that the MiSVP2 protein may have a similar function to the PvSVP protein. The genome sequence of ‘Alfonso’ mango has been published; although there are two base differences between the *SVP1* gene of ‘SiJiMi’ and the *SVP1* gene of ‘Alfonso’, there is no difference in amino acid sequence. The sequence of the *SVP2* gene in ‘SiJiMi’ is the same as that in ‘Alfonso’.

In *A. thaliana*, *AtSVP* is mainly expressed in vegetative tissues but not in flowers and pods [9]. Strawberry *FaSVPs* and kiwifruit *AcSVPs* are expressed only in vegetative organs and not in reproductive organs, which is consistent with the patterns in *A. thaliana* [24,25,26]. However, some different expression patterns have been found; for example, for *Medicago MtSVP*, *Chrysanthemum morifolium CmSVP*, grape lily *LoSVP*, *Eriobotrya japonica EjSVPs*, *Prunus mume PmSVPSs*, *Malus × domestica MdSVPs*, and another apple *SVP*-like gene, *MdMADS50*, are all of highly expressed in vegetative tissues but weakly expressed in reproductive organs [23,27,28,29,30,31,32]. In this study, the qRT-PCR results suggested that the expression patterns of *MiSVP1* and *MiSVP2* were the same; however, the transcription levels of *MiSVP1* and *MiSVP2* were different. *MiSVP1* and *MiSVP2* were highly expressed in vegetative tissues, but their expression levels were low in flowers/buds. These findings suggest that the expression and function of SVP homologous proteins may be different in different species.

The temporal expression patterns indicated that the expression of *MiSVP1* in buds increased during the vegetative growth period but decreased during flower development. This result is consistent with the expression patterns of *SVP*-like genes in other species, such as *Poncirus trifoliata L.* Raf., *Cymbidium goeringii*, and *Crocus sativus* L. [33,34,35]. However, the expression of *MiSVP2* decreased during the vegetative stage and flower induction stage and increased during flower development. These results suggest that the *SVP* gene may have evolved different temporal expression patterns, a possibility that needs further study.

The *MiSVPs* showed sensitivity to low temperature, PEG and NaCl treatments, especially low temperature; PEG treatment significantly promoted their expression. The *SVP* gene is a key factor in the response to low temperature [13]. In *A. thaliana*, lily, and *Cymbidium goeringii*, low temperature can promote the expression of *SVP* genes, which agrees with the results of the present study [28,34,36]. In previous work, 20% PEG and 150 mM NaCl treatment did not significantly change the expression levels of *SVPs* in *A. thaliana*, which differs from the results of the present study [36]. The expression level of *MiSVPs* in mango seedlings changed significantly under treatment with 30% PEG and 300 mM NaCl. PEG treatment can significantly promote the expression of *MiSVPs.* After NaCl treatment, the expression of *MiSVP1* was significantly downregulated at 6 h, upregulated to a peak at 12 h, and then downregulated again. The expression of *MiSVP2* was significantly downregulated at 6 h, restored to the pretreatment level at 12–24 h, and significantly increased at 48–72 h, indicating that the expression of *MiSVPs* is regulated by PEG and NaCl. The reason for the different results between studies may be that the concentrations of PEG and NaCl used in *Arabidopsis* were different from those in this study.

*SVP* homologous genes in most species mainly inhibit flowering, participate in the development of floral organs and maintain flower meristem specificity. In *Arabidopsis*, the *SVP* mutant shows early flowering and a reduced number of rosette leaves and stem leaves [9]. Loquat *EjSVP1* and *EjSVP2* and *Brassica BcSVP* have been found to delay flowering and induce floral organ variation [30,37]. Overexpression of *OsMADS22*, *OsMADS47* and *OsMADS55* in rice can delay flowering, but only the transgenic plants of *OsMADS22* and *OsMADS47* have been found to show floral organ variation [38]. Overexpression of *Prunus mume PmSVP1* and *PmSVP2* showed floral organ variation, but only *PmSVP1* delays flowering [23]. Overexpression of four *kiwifruit AcSVP* genes results in floral organ variation, but only *AcSVP1* delays flowering [26,39,40]. In addition, *SVP* genes play a role in promoting flowering in some species. For example, overexpression of *Epimedium sagittatum EsSVP* can produce early flowering lines and late flowering lines [41]. Overexpression of the *Polypogon fugax SVP* homologous gene *PfMADS16* leads to early flowering in transgenic plants [42]. Overexpression of *Phyllostachys violascens PvSVP1* and *PvSVP2* can induce early flowering and floral organ variation [43]. In this study, overexpression of *MiSVP1* and *MiSVP2* had no effect on the inflorescence structure or the number of rosette leaves in transgenic plants. *MiSVP1* delayed flowering, and *MiSVP2* resulted in early flowering. These results suggest that *SVP* genes play different roles in different species and that different *SVP* homologous genes in the same species play different roles in plant growth.

In *A. thaliana*, SVP and FLC can interact with each other to form a heterodimer, SVP-FLC, which can be combined with the promoter regions of SOC1 and FT to regulate the expression of SOC1 and FT, thereby regulating the flowering in *A. thaliana* [12,25]. Research has found that SVP can interact with AP1 and participate in the regulation of the flowering stage [42]. In this study, overexpression of the *MiSVP* gene in *A. thaliana* caused no significant changes in the number of rosette leaves; however, it had a significant impact on flowering time. *MiSVP1* can significantly delay the flowering time of transgenic plants compared to that of WTplant, under *MiSVP1* overexpression, the expression of endogenous genes *AtFT* and *AtSOC1* in *Arabidopsis* decreased, while the expression of the endogenous gene *AtFLC* increased, which is consistent with the effects of *PmSVP2* [23]. *MiSVP2* significantly accelerated the flowering time of transgenic plants; additionally, the expression levels of endogenous genes exhibited opposite changes to those under *MiSVP1*: the expression levels of the endogenous genes *AtFT* and *AtSOC1* increased, while the expression of the endogenous gene *AtFLC* decreased, consistent with the effects of *SVP*-like gene *PfMADS16* [42]. In conclusion, *MiSVPs* regulate flowering time in transgenic plants by regulating the expression levels of the *FT* and *SOC1* genes.

Among the four *SVP*-like genes of kiwifruit, SVP1 and SVP4 can interact with SOC1 but not with AP1 and FLC, and SVP2 and SVP3 can interact with AP1, SOC1, and FLC, playing a redundant regulatory role [25,26]. In pecan, CcSVP can interact with CcAP1 and CcSOC1 but not with CcFLC [44]. In rice, OsMADS22 and OsMADS55, which are encoded by SVP homologous genes, interact with AGL24 and AP1, while only OsMADS55 interacts with FLC [38]. In *Cymbidium goeringii*, CgSVP can interact with CgAP1 and CgSOC1 [34]. These observations indicate that the numbers and regulatory mechanisms of *SVP* genes differ among different plants, suggesting that the *SVP* homologous genes from different species have evolved different functions. In this study, MiSVP1 and MiSVP2 were found able to interact with SEP1-1, AP1-2, and SOC1D. Our results are similar to those of previous studies.

## 4. Materials and Methods

### 4.1. Plant Materials

The test materials used in this study were from 10-month-old and 5-year-old ‘SiJiMi’ mango (*Mangifera indica* L.) trees growing in the Fruit Tree Specimen Garden of the Agricultural College of Guangxi University (Nanning, China). The experimental materials for cloning from the leaves of ‘SiJiMi’ mainly included healthy young stems, flowers, leaves of 5-year-old seedlings and leaves of 10-month-old seedlings of ‘SiJiMi’. Leaves of ‘SiJiMi’ were collected once a month starting in November 2018 from flower bud differentiation to the flowering stage, for a total of five times. Mature leaves of 10-month-old seedlings of ‘SiJiMi’ were treated with 300 mM NaCl, 30% PEG or 2 °C, and leaves were collected at 0, 6, 12, 24, 48, and 72 h after treatment to analyze the effects of NaCl, PEG, or 2 °C treatment on the expression levels of *MiSVPs*. The samples used in the experiment were all collected from the healthy leaves of the fourth round from the top bud on the new branches; the flowers were all unopened or half-opened. The stems were collected from the new branches of the mango in the same year, and the buds were the top buds on the mango branches. All samples were stored in a refrigerator at −80 °C to prevent RNA degradation.

Wild-type *A. thaliana* plant were preserved in our laboratory only as transgenic receptors. *Nicotiana* tobacco was preserved in our laboratory only as a receptor for subcellular localization experiments. *Arabidopsis* and tobacco were cultured at 22 °C under for a long photoperiod (16 h light/8 h dark cycle).

### 4.2. Cloning and Bioinformatics Analysis of MiSVPs

The total RNA extraction kit of TIGANDAN DP441 polysaccharide polyphenol plants was used to extract the total RNA of mango, and Moloney mouse leukemia virus (M-MLV) reverse transcriptase (TaKaRa, Dalian, China) and primer AUP1 were used to perform reverse transcription following the manufacturer’s instructions to obtain cDNA. The specific primers MiSVP1/2u and MiSVP1d, MiSVP2d were designed, and PCR with cDNA as a template was performed to obtain MiSVP1 and MiSVP2 fragments (Appendix A).

Identification of *MiSVP* nucleotide sequences was completed through the National Center for Biotechnology Information (NCBI) BLAST program (http://www.ncbi.nlm.nih.gov/BLAST, accessed on 1 June 2018). Exon–intron structures of *MiSVP1* and *MiSVP2* were generated using IBS 1.0 [45]. The aa sequence, MW and pI of the MiSVP proteins were determined via BioXM 2.6 software (Ji Huang, Nanjing, China). Multiple alignments of the two *MiSVP* genes and *SVP*-like aa sequences in different species were performed using DNAMAN software (Lynnon Biosoft, Vaudreuil-Dorion, Quebec, Canada). The phylogenetic tree of the two MiSVP and SVP-like proteins was constructed via MEGA 6.0 software (Koichiro Tamura, Hachioji, Tokyo, Japan), with 1000 bootstrap replicates. SVP-like aa sequences in different species were downloaded from NCBI.

### 4.3. Expression Analysis of MiSVPs

The qRT-PCR-specific primers for *MiSVP1* and *MiSVP2* were designed in Primer3 Input (https://bioinfo.ut.ee/primer3-0.4.0/, accessed on 13 January 2019) (Appendix A). The expression levels of *MiSVP1* and *MiSVP2* were analyzed on an ABI 7500 instrument (Applied Biosystems, Foster, America) by qRT-PCR. The qRT-PCR conditions followed the manufacturer’s instructions. Each qRT-PCR mixture contained 10 µL SYBR Premix Ex Taq II (TakaRa, Dalian, China), 0.5 µL (10 µM) upstream and downstream specific primers of *MiSVPs* (Appendix A), 0.4 µL ROX reference dye II and 2 µL cDNA (100 ng·µL^−1^) as template, and sterile water was added to bring the total system volume to 20 µL. The mango *MiACT1* gene was used as the internal reference gene [46], and the relative expression levels of *MiSVPs* were calculated by the 2^−ΔΔCt^ method [47].

### 4.4. Subcellular Localization

The *MiSVP* gene and the vector p1300 plasmid were connected through *SacI* and *BamHI* cleavage sites, and the vector p1300-MiSVP1-GFP and p1300-MiSVP2-GFP fusion protein carriers were constructed and transferred into EHA105 cells. EHA105 cells, which contained plasmids carrying the target gene, were incubated overnight in 1 mL YEP liquid culture medium (50 µL/100 mL KANA, 25 µL/100 mL Rif) at 28 °C and oscillated at 200 rpm. The bacteria were collected by centrifugation (8000 rpm, 5 min), resuspended to OD600 = 0.6–1.0, transferred into tobacco by the EHA105-mediated method, cultured at 25 °C for 2 days, and then observed through confocal laser scanning microscopy (TCS-SP8 MP, Leica, Germany).

### 4.5. MiSVP Vector Construction and Transformation of Arabidopsis

The pBI121-GUS-*MiSVP1* and pBI121-GUS-*MiSVP2* overexpression vectors were constructed and transformed into EHA105 cells. WT *A. thaliana* was transformed by the floral dipping method with *Agrobacterium tumefaciens* containing the target gene [48]. Positive seedlings (T1) were obtained by resistance screening on 1/2 MS solid medium supplemented with kanamycin (100 mg/L). The leaves of T1 generation transgenic plants were collected to extract DNA by the CTAB method, and PCR was used to detect whether the T1 generation plants contained target genes. Seeds of T1 generation plants were further screened in the resistant medium. The positive seedlings with the ratio of positive seedlings to negative seedlings of 3:1 in the resistant medium were transplanted, then T2 generation plants were obtained. The leaves of T2 generation transgenic plants were collected for RNA extraction. The expression levels of *MiSVPs* and original genes in WT and transgenic plants were analyzed by qRT-PCR, and the primers used in this study are shown in Appendix A. Seeds of T2 generation plants were collected and further screened in the resistant medium. If all the seedlings survive in the resistant medium, it is the T3 homozygous transgenic line. Phenotypic analysis was performed with the WT, pBI121-GUS and T3 homozygous transgenic lines. The number of rosette leaves was recorded when the bolting height of the plant was 0.5–1 cm. The days to initial flowering was calculated as the period from sowing to the first bloom. The leaves of WT plants and transgenic plants employed for detection were collected on the same day before flowering to extract total RNA by the Easy Pure^®®^ Plant RNA Kit (Transgen Biotech, Being, China) following the manufacturer’s instructions. With *Arabidopsis AtACTIN* as an internal reference gene (Appendix A), the relative expression levels of the *MiSVP* genes and endogenous genes were calculated by the 2^−ΔΔCt^ method [46,47].

### 4.6. Yeast Two-Hybrid Assay

The pGBKT7-MiSVP1 and pGBKT7-MiSVP2 fusion protein vectors were constructed and transferred into Y2H Gold competent cells. Specific primers were used for PCR detection to confirm successful transformation, and then self-activation and toxicity tests were performed (Appendix A). Protein–protein interaction screening was performed using pGBKT7- MiSVPs and the ‘SiJiMi’ mango cDNA library following the manufacturer’s instructions (Clontech). The incubated mixed products were initially screened in SDO/–Trp/–Leu/–His/–Ade (QDO). The products of preliminary screening were further screened on SDO/–Trp/–Leu/–His/–Ade/X-alpha-gal/AbA (QDO/X/A) solid medium. The common primers T7 and 3′AD were used to detect the obtained blue monocolonies (Appendix A). The plasmid of the candidate protein was extracted and retransformed into the competent Y187 yeast cells. The pGBKT7-T was used to detect the self-activation of candidate proteins. A point-to-point assay was performed to verify the interaction between nonself-activated candidate proteins and MiSVPs.

### 4.7. Bimolecular Fluorescence Complementation (BiFC) Assay

The full-length sequence of the MiSVP protein was constructed into puc-sPYNE, the full-length sequence of the candidate protein was constructed into puc-sPYCE, and the fusion vector was transferred into EHA105. *Agrobacterium*-mediated technology was used to transfer *Agrobacterium* containing the fusion vector into onion epidermis, which was then observed through confocal laser scanning microscopy (TCS-SP8MP, Leica, Germany).

### 4.8. Statistical Analysis

IBM SPSS 19.0 (SPSS Inc., Chicago, IL, United States) was used to analyze the experimental data, Duncan’s multiple range test was used to determine the statistical significance of the results, and *p* < 0.05 indicated a significant difference.

## 5. Conclusions

In this study, two *MiSVP* genes were isolated and identified from mango. The temporal and tissue expression analysis and the results of low temperature, PEG and NaCl treatment showed that the *MiSVPs* were expressed in various stages of flower development, mainly in vegetative tissues, and that *MiSVPs* can respond to plant regulation of low temperature, drought, and salt stress. Furthermore, the overexpression of *MiSVPs* had no significant effects on the number of rosette leaves; however, it had a significant impact on flowering time. Point-to-point and BiFC assays showed that the two MiSVP genes could interact with the flowering-related proteins AP1-2, SEP1-1, and SOC1D. These findings provide important insights into the regulation of the mango flowering stage; however, the main reason why the functions of the two *MiSVP* genes differ in regulating flowering time needs further investigation.

## Figures and Tables

**Figure 1 ijms-22-09802-f001:**
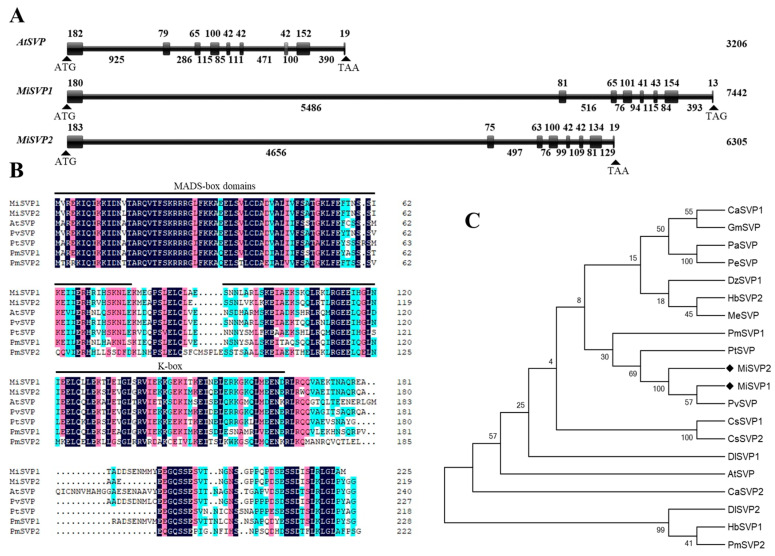
Sequence, structural, and phylogenetic analyses of MiSVP1 and MiSVP2. (**A**) Schematic diagram of intron/exon structures of *MiSVP1, MiSVP2*, and *AtSVP*. The lines represent introns, and the boxes represent exons. The number represents the length of the exons/introns in base pairs. (**B**) Multiple amino acid sequence alignment of MiSVPs and SVP-like proteins in different species. The underscores denote the MADS-box and K-box domains. Black indicates that the sequences are identical, red indicates ≥ 75% similarity, and blue indicates ≥ 50% similarity. (**C**) Phylogenetic analysis of SVP proteins from mango and other species. MiSVP1 and MiSVP2 are marked by black diamonds. The protein sequences used in this study were downloaded from NCBI, and were as follows (gene accession numbers in parentheses): CaSVP1 (*Coffea arabica*, AHW58026), CaSVP2 (*Coffea arabica*, AHW58042), GmSVP (*Glycine max*, NP_001240951), PaSVP (*Populus alba*, XP_034933033), PeSVP (*Populus euphratica*, XP_011021845), DzSVP1 (*Durio zibethinus*, XP_022751126), HbSVP1 (*Hevea brasiliensis*, ARQ16479), HbSVP2 (*Hevea brasiliensis*, ARQ16481), MeSVP (*Manihot esculenta*, XP_021631111), PmSVP1 (*Prunus mume*, AML81015), PmSVP2 (*Prunus mume*, AML81016), PtSVP (*Citrus trifoliata*, ACJ09169), PvSVP (*Pistacia vera*, XP_031287661), CsSVP1 (*Camellia sinensis*, XP_028064640), CsSVP2 (*Camellia sinensis*, XP_028064641), DlSVP1 (*Dimocarpus longan*, AIY25020), DlSVP2 (*Dimocarpus longan*, AIY25021), AtSVP (*Arabidopsis thaliana*, AF211171).

**Figure 2 ijms-22-09802-f002:**
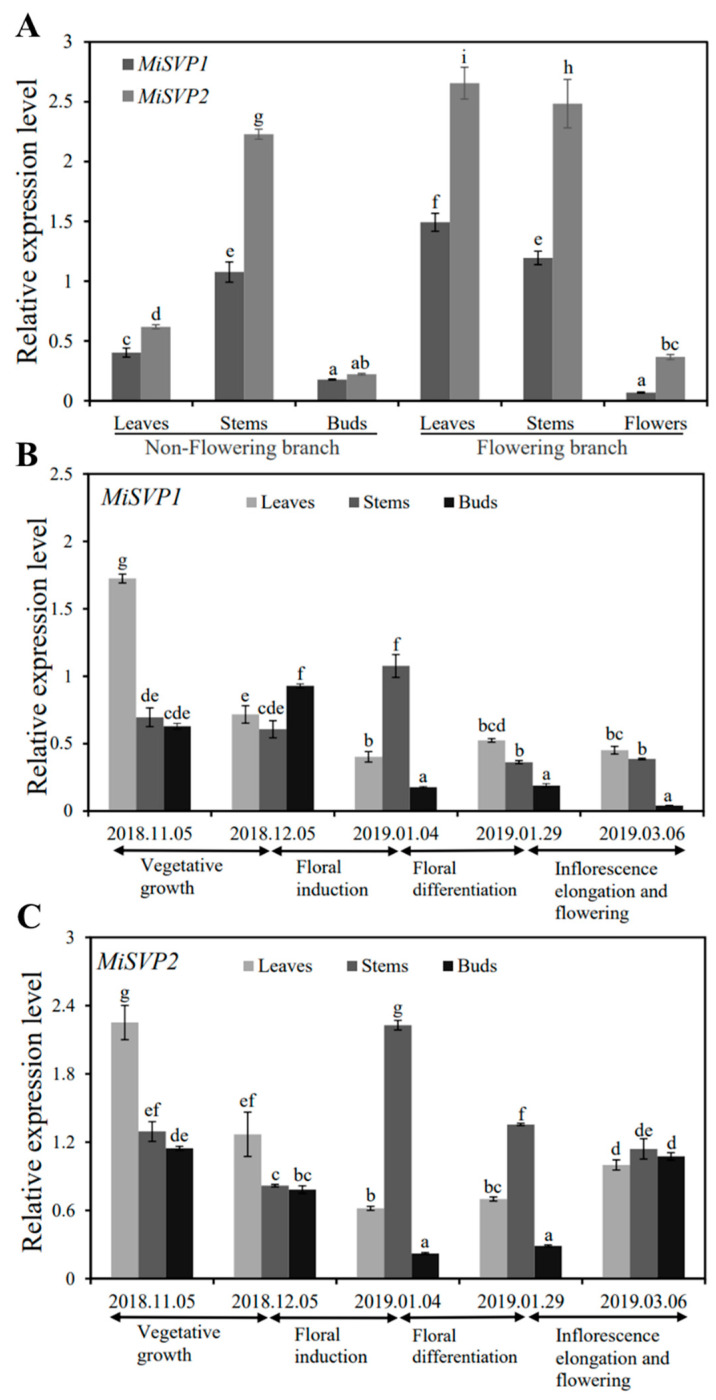
Tissue and temporal expression analyses of *MiSVP1* and *MiSVP2* in mango. (**A**) Tissue expression analysis of *MiSVP1* and *MiSVP2* in the different tissues of mango. (**B**,**C**) Temporal expression analysis of *MiSVP1* and *MiSVP2* at different floral development stages. The vegetative growth phase was 5 November 2018, to 5 December 2018. Floral induction was 5 December 2018 to 4 January 2019. The floral differentiation phase was 4 January 2019 to 29 January 2019. The inflorescence elongation and flowering phase was 29 January 2019 to 6 March 2019. The *MiActin1* gene served as an internal control. Duncan’s multiple range test was used to analyze the significance of differences among multiple groups of samples, and *p* < 0.05 indicated a significant difference. The error bars represents the standard error of the mean (±SE) among three replicates of the same sample.

**Figure 3 ijms-22-09802-f003:**
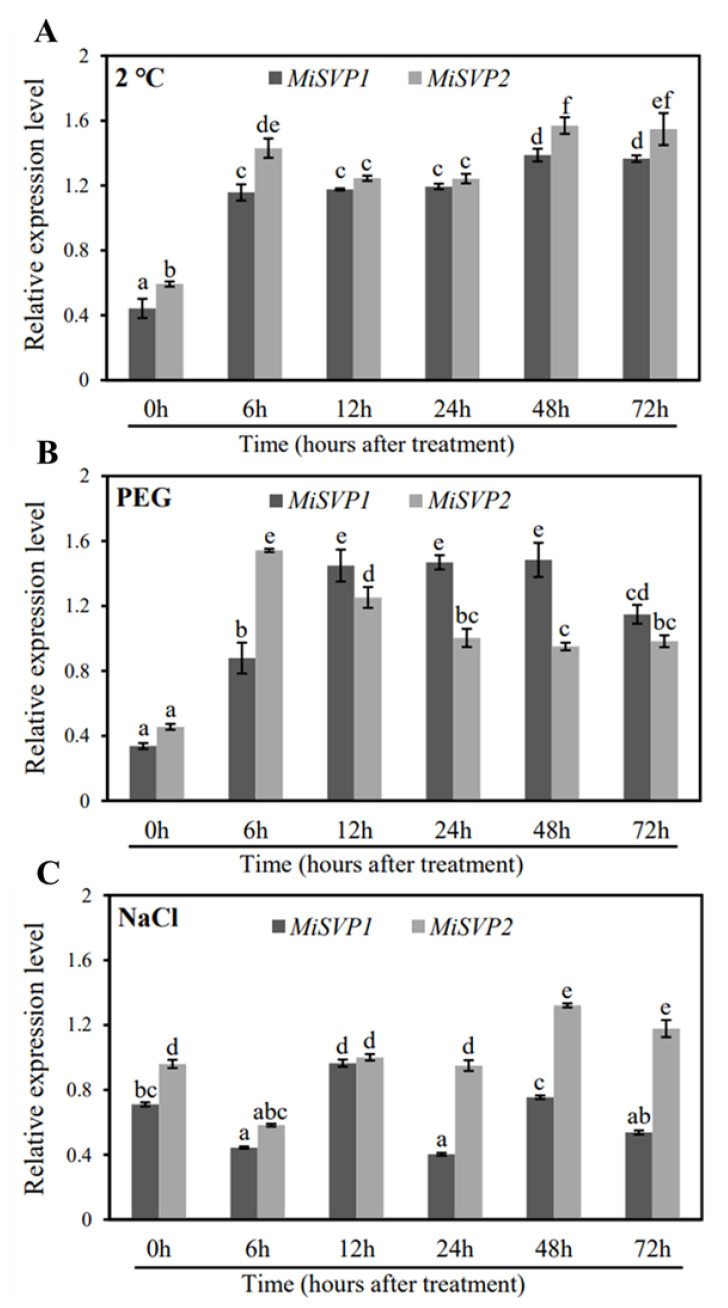
Expression patterns of *MiSVP1* and *MiSVP2* in leaves under stress treatments. (**A**) 2 °C (low temperature) treatment. (**B**) PEG treatment. (**C**) NaCl treatment. Duncan’s multiple range test was used to analyze the significance of differences among multiple groups of samples, and *p* < 0.05 indicated a significant difference. The error bars represent the standard error of the mean (±SE) among three replicates of the same sample.

**Figure 4 ijms-22-09802-f004:**
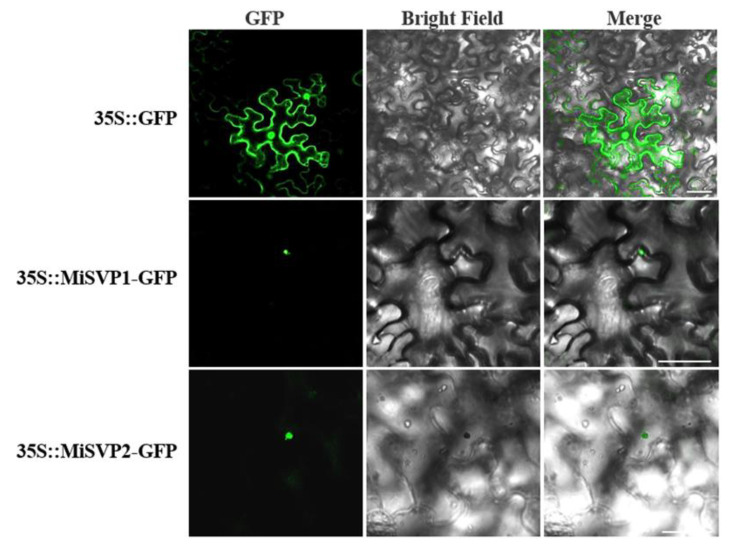
Subcellular localization of MiSVPs in tobacco leaves. GFP: Signaling of a GFP-fused protein by fluorescence microscopy; Merged: superimposed GFP and bright-field images. Bars = 50 μm.

**Figure 5 ijms-22-09802-f005:**
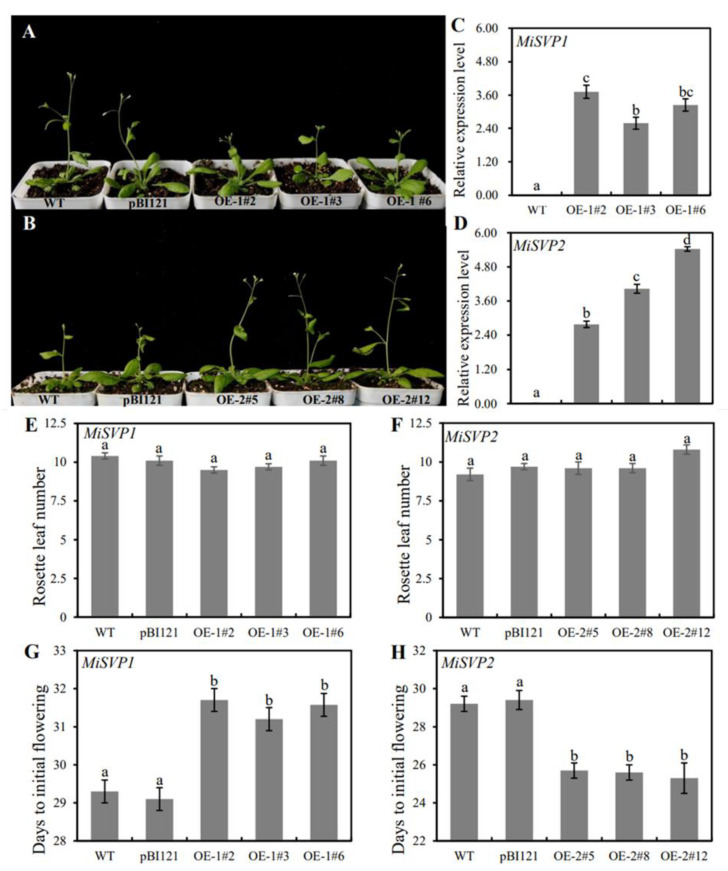
Effects of overexpression of *MiSVP1* and *MiSVP2* on the phenotype of *Arabidopsis.* (**A**) Phenotypes of *MiSVP1* transgenic lines and controls. (**B**) Phenotypes of *MiSVP2* transgenic lines and controls. The controls were WT and pBI121-GUS plants. (**C**,**D**) The expression levels of *MiSVP1* and *MiSVP2* in transgenic plants. (**E**,**F**) The number of rosette leaves of *MiSVP1* and *MiSVP2* transgenic lines and the WT. (**G**,**H**) The days to initial flowering of *MiSVP1* and *MiSVP2* transgenic lines and the WT. Duncan’s multiple range test was used to analyze the significance of differences among multiple groups of samples, and *p* < 0.05 indicated a significant difference. The error bars represent the standard error of the mean (±SE) among 12 replicates of the same sample.

**Figure 6 ijms-22-09802-f006:**
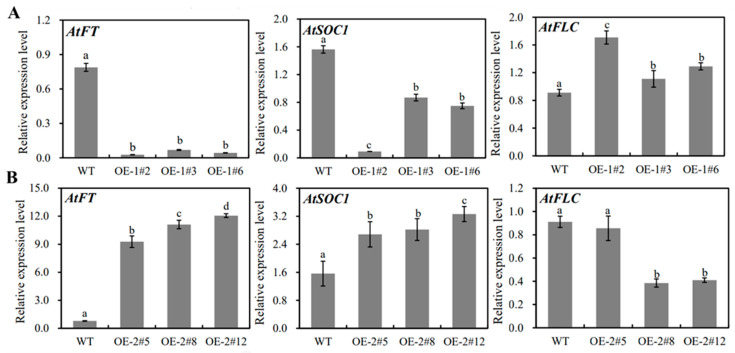
Expression level analysis of endogenous genes in *Arabidopsis*. (**A**) Expression of *AtFT, AtSOC1*, and *AtFLC* in 35S::*MiSVP1* transgenic plants. (**B**) Expression of *AtFT*, *AtSOC1*, and *AtFLC* in 35S::*MiSVP2* transgenic plants. Relative expression level was determined by comparing the expression levels of *AtFT*, *AtSOC1*, and *AtFLC* with the expression level of *AtACTIN*. Duncan’s multiple range test was used to analyze the significance of differences among multiple groups of samples, and *p* < 0.05 indicated a significant difference. The error bars represent the standard error of the mean (±SE) among three replicates of the same sample.

**Figure 7 ijms-22-09802-f007:**
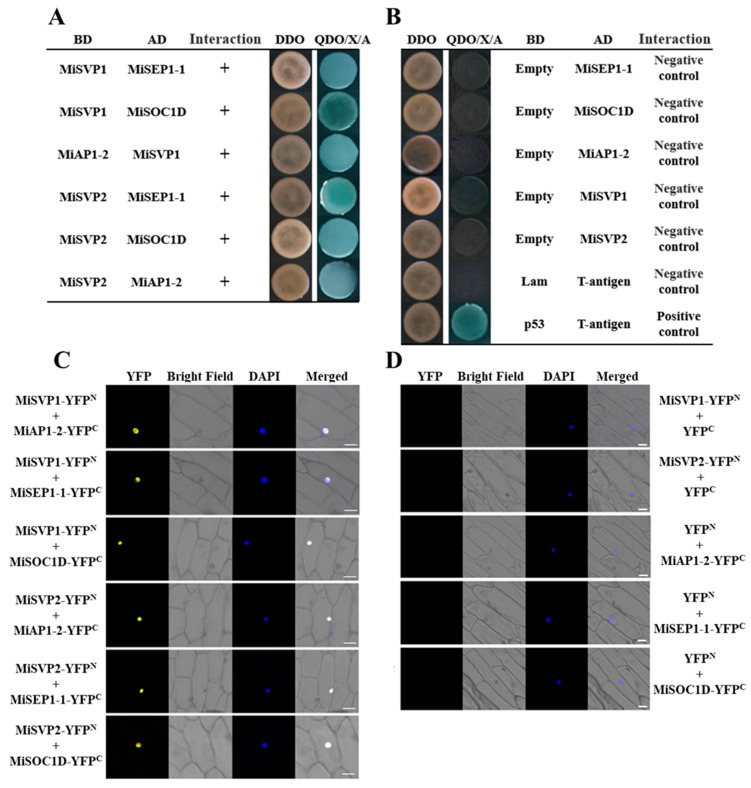
Identification of MiSVP1 and MiSVP2 interacting proteins. (**A**) Point-to-point verification of MiSVP1and MiSVP2 with candidate proteins. (**B**) The self-activation verification of candidate proteins. (**C**) BiFC assays of MiSVP1and MiSVP2 with candidate proteins. (**D**) BiFC assays of MiSVPs and candidate proteins with empty vector. YFP: Signaling of a YFP-fused protein by fluorescence microscopy; DAPI: staining shows cell nucleus; Merged: superimposed YFP, DAPI, and bright-field images. Bars = 20 μm.

## Data Availability

Not applicable.

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
