# Peer review of "Isolation and Functional Characterization of Two *SHORT VEGETATIVE PHASE* Homologous Genes from Mango"

_ijms, 2021, doi:10.3390/ijms22189802_

Round 1

Reviewer 1 Report

This study isolated two SVP-like genes from mango and analyzed their characteristics and functions. There are no difficulties in the experimental methods or results, and I believe that this manuscript contains contents worthy of publication in IJMS. Please refer to the following comments for correction.

  1. The early or delayed flowering in Arabidopsis plants overexpressed MiSVP1 and MiSVP2 suggests that MiSVP1 and MiSVP2 have opposite effects. However, the expression pattern in mango does not suggest that these genes have opposite effects on mango flowering. Also, the results of the two-hybrid assay do not show that these genes have opposite effects, but rather suggest that they bind to the same proteins and have the same functions. What do you think about this point?
  2. It is known that flowering of mangoes is induced by cool temperature. The period from December to January, which is the induction period of flowering, is the time when the temperature drops. During this period, both MiSVP1 and MiSVP2 are up-regulated in branches, but down-regulated in leaves and buds. This result seems to contradict the fact that the expression of MiSVP1 and MiSVP2 is up-regulated by low temperature in Fig. 3. What do you think about this point?

L275-276  Isn't it Arabidopsis that transformed with PmSVP in the study?

L302-304   This sentence is not mentioning about mango, but overexpressed Arabidopsis. Please make it clear.

Reviewer 2 Report

This research article is devoted to the study of two flowering-related SVP genes in mango. Homologs of these genes have been previously analyzed in many other plant species. Authors have demonstrated the tissue-specific expression of the genes and studied the effect of several environmental factors on the expression of these genes. The functions of the genes were demonstrated through overexpression in the Arabidopsis plants.
The novelty of the presented data is not obvious. 
There are few flaws in the manuscript. Only one reference gene was used for the normalization of qPCR data. Normally, the data have to be normalized using at least two reference genes.
The images in the Figure 7 A and B have to be replaced with other pictures. All samples showing protein-protein interaction, including a positive control, have a different bluish background, while in all samples with no interaction – the background is of a different color. This casts doubt on the reliability of the data presented.
Line 34, please, clarify what does “autonomous” factors mean in this sentence. 
In the Introduction section, a better-structured description of the regulation of flowering time and flower development is needed.
Line 43-44 In the Introduction of this manuscript, describing previous publication, the authors wrote: “SVP antagonism with SOC1 in the meristem represses TWIN SISTER OF FT (TSF) in vascular tissue of leaves [10]”, In the original article it is written: “Thus we distinguish the functions of SVP in repressing FT and TSF in the leaf and SOC1 in the meristem”. Please, correct the description of the work of ref. 10.
Please, give your explanation/suggestion as, why MiSVP2 expression increases in buds during flowering.
Fig. 2 B, C – arrows /lines under the graphs are shifted, according to the figure legend.
Line 112 and other places in the text – The term “Prophase” describes the cell cycle phase. If possible, please, substitute the term “prophase” with another word (example “phase”).
Line 129 – should be “the alteration of the expression level”
Lines 157-160 It is written “Three homozygous transgenic lines of MiSVP1 (OE-1#2, OE-1#3 and OE-1#6) and MiSVP2 (OE-2#5, OE-2#8 and OE-2#12) were chosen for phenotype analysis. Three homozygous transgenic lines of MiSVP1 (OE-1#2, OE-1#3 and OE-1#6) and MiSVP2 (OE-2#5, OE-2#8 and OE-2#12) were selected for functional analysis.” Please, combine these two sentences in one. 
Data presented in Fig. 5 (G, H): For the accurate estimation of the flowering time the analysis of a bigger number of plants is needed, but according to the figure legend: “The error bar represents the standard error of the mean (±SE) between three replicates of the same sample”.
Line 324 “300 mmol/L NaCl”. Please, substitute with 300 mM NaCl.
Line 326 “The leaves used in the experiment were all collected from the healthy leaves of the fourth round from the top bud…” Leaves collected from leaves? Please, correct the sentence.
Paragraph 4.5 of Material and Methods section: According to the description, the homozygosity of the transgenic plants was not confirmed. Please, explain, what does “further screening” mean (lines 382-383) 
There are multiple grammatical mistakes.

Reviewer 3 Report

The authors have obtained two full-length cDNA sequences of SVP homologous genes (MiSVP1 and 11 MiSVP2) from ‘SiJiMi’ mango. They analyzed MiSVPs sequence analysis and the expression patterns in different tissues and during flower development stages in leaves, stems and buds by RT-qPCR analysis. Furthermore, they evaluated the MiSVPs under three abiotic stresses (low temperature, NaCl and PEG). In this study, the Authors have revealed the subcellular localization of MiSVP1 and MiSVP2 and the functional analysis of MiSVPs in Arabidopsis thaliana.

The manuscript is well-written in an unambiguous and technically correct style.

Introduction is well-structured. The background of your study provide context to the information discussed throughout this paper.

M&M are well-described and structured, with high technical standard.

In the paper all results are relevant to the hypothesis and are sufficiently supported by discussions

Generally, the paper contains some interesting data, and, thus, yields new information in this research area.

However, the paper should be revised as recommended, and some issues should be changed (see file for the Authors).

Round 2

Reviewer 2 Report

The manuscript is improved after the revision.

A couple of minor comments:

Please, introduce the information about the selection of homozygous transgenic lines (response to Point 13) to the Material and Method section.

Line 34-36 Reviewer still suggests including further explanation about what “autonomous pathway” means. Example: “the autonomous pathway comprising a combination of epigenetic factors and post-transcriptional gene regulation”.
